# Neuroprotective Norsesquiterpenoids and Triterpenoids from *Populus euphratica* Resins

**DOI:** 10.3390/molecules24234379

**Published:** 2019-11-30

**Authors:** Yun-Yun Liu, Dan-Ling Huang, Yun Dong, Da-Peng Qin, Yong-Ming Yan, Yong-Xian Cheng

**Affiliations:** 1School of Pharmaceutical Sciences, Shenzhen University Health Science Center, Shenzhen 518060, China; liuyunyun2017@foxmail.com (Y.-Y.L.); leonchemistry@szu.edu.cn (D.-L.H.); yundong@szu.edu.cn (Y.D.); tqindp@szu.edu.cn (D.-P.Q.); yanym@szu.edu.cn (Y.-M.Y.); 2Department of Chinese Materia Medica, Shanxi University of Chinese Medicine, Jinzhong 030619, China

**Keywords:** *Populus euphratica*, plant resins, sesquiterpenoids, triterpenoids, neuroprotection

## Abstract

Two new octanorlanostane-type triterpenes, euphraticanoids A and B (**1** and **2**), two new trinorsesquiterpenoids, euphraticanoids C and D (**3** and **4**), and eight known triterpenoids (**5**, **6**, **8**–**13**) along with one steroid (**7**) were isolated from *Populus euphratica* resins. The structures of these new compounds, including their absolute configurations, were characterized by spectrocsopic, chemical, and computational methods. Biological evaluation revealed that compounds **4**, **7**–**9**, **12**, and **13** display neuroprotective activities in H_2_O_2_-induced HT-22 cells with **4**, **8**, and **9** occurring in a concentration-dependent manner and **7**, **12**, and **13** reaching the maximum effects at 20 μM. Meanwhile, the neuroprotective properties of all isolates were accessed using glutamate-induced SH-SY5Y cells and disclosed that compounds **3**, **4**, **8**, and **9** could dose-dependently protect neural cell injury in a concentration range of 10–40 μM. Finally, a brief structure–activity relationship was briefly discussed.

## 1. Introduction

*Populus euphratica*, a plant of the family Salicaceae, spreads over the world in places such as China, Russia, Mongolia, India, and Iran. In China, the tree is mainly distributed in the west of China with abiotic surroundings exemplified as desert or saline and alkaline lands [1]. The resins secreted by the tree, known by the elegant name “the tears of poplar,” have been used to treat tuberculous adenitis, throat, and duodenal ulcer swelling in China [2]. Previous studies revealed the presence of salicin derivatives, volatile oils, and phenolics in the resins of *P. euphratica* [3,4]. In recent years, we have become interested in chemical investigations of medicinal plant resins. As a result, an increasing number of structurally novel terpenoids have been characterized [5,6,7]. In a continuous study on medicinal resins, the title material was investigated, resulting in the isolation of diterpenoids with cytotoxic and potent wound-healing promotion properties [8,9,10]. The current work is an in-depth investigation on *P. euphratica* resins, which led to the characterization of 12 terpenoids and one steroid with euphraticanoid compounds A–D (**1**–**4**) being new ones (Figure 1). To get an insight into the biological profiling of these secondary metabolites, neuroprotective properties of all the isolates were evaluated in either H_2_O_2_ or glutamate-induced neural or human neuroblastoma cells. In this paper, we describe the isolation, structure characterization, and neuroprotective activities of all the isolated compounds.

## 2. Results and Discussion

### 2.1. Structure Elucidation of the Compounds

Compound **1** was obtained as a white powder with a positive optical rotation ([*α*]_D_^25^ +2.04; in MeOH). Its molecular formula was deduced as C_22_H_34_O on the basis of its HRESIMS (high resolution electrospray ionization mass spectroscopy), ^13^C NMR, and DEPT spectra. The ^1^H NMR spectrum of **1** shows signals for five methyl singlets (*δ*_H_ 1.05, 0.98, 0.84, 0.84, and 0.64). The ^13^C NMR and DEPT spectra (Table 1) indicate 22 carbons ascribed to five methyl, six methylene, six methine with three sp^2^ ones, and five non-protonated carbons (four aliphatic and one olefinic). A comparison NMR data of **1** with those of commiphorane G2 [11] revealed their resemblance except for a ^13^C NMR chemical shift difference occurring at C-1, C-2, C-3, and C-4. The ^1^H-^1^H COSY spectrum (Figure 2) shows correlations of H-1/H-2/H-3 (*δ*_H_ 3.42), in combination with the chemical shift of C-3 (*δ*_C_ 76.3), suggesting the chemical shift alterations in ring A (Figure 2) between **1** and commiphorane G2 might result from the configuration at C-3. To confirm this conclusion, a ROESY experiment (Figure 3) was utilized, and the ROESY cross peaks of Ha-2/H_3_-19, H_3_-21, H_3_-19/H_3_-18 indicate these protons are adjacent in space and assigned at *β*-orientation. Further, cross peaks of H-5/Ha-7, H_3_-20; H_3_-22/Ha-7, Hb-12, Hb-12/H-9, and H-9/Hb-2 suggest that these protons are at *α*-oriented. Finally, we examined the relative configuration of C-3 using the nuclear Overhauser effect (NOE) irradiation. A NOE enhancement was observed between 3-OH and H_3_-20 (Appendix A), evidently indicating the *α*-orientation of 3-OH in **1** contrary to that of commiphorane G2. Further NMR data comparison of ring A of **1** with the counterpart of epimansumbinol (**6**) [12] which bears a 3*α*-OH found their accordance, also securing the above conclusion. Thus, the planar structure and the relative configuration of **1** was deduced as shown in Figure 1. To clarify the absolute configuration of **1**, electronic circular dichroism (ECD) calculation was carried out at B3LYP/6-311 + G(d) l level. The results show that the calculated ECD spectrum of (3*R*,5*R*,8*R*,9*R*,10*R*,14*R*)-**1** (Figure 4) agrees well with the experimental one, indicating the absolute configuration of **1** is 3*R*,5*R*,8*R*,9*R*,10*R*,14*R*. In this way, the structure of **1** was identified and named as euphraticanoid A.

Compound **2** was isolated as a white powder with a positive optical rotation ([*α*]_D_^25^ +7.27, in MeOH). The molecular formula of **2** was assigned as C_22_H_34_O aided with its HRESIMS, ^13^C NMR, and DEPT spectra. The ^1^H NMR spectrum of **2** exhibits five methyl (*δ*_H_ 1.08, 0.97, 0.93, 0.86, and 0.71). The ^13^C NMR and DEPT spectra (Table 1) display 22 carbons classified into five methyl, six methylene, six methine (three sp^2^), and five non-protonated carbons. These NMR signals resemble those of **1**, indicating they are analogs. Compound **2** differs from **1** only in the position of two double bonds. The *Δ*^12(13)^ and *Δ*^16(17)^ double bonds in **2** rather than the *Δ*^13(17)^ and *Δ*^15(16)^ ones in **1** were observed to be supported by the ^1^H-^1^H COSY correlations of H-11/H-12 (*δ*_H_ 5.40) and H-15 (*δ*_H_ 2.59, 1.85)/H-16/H-17, and HMBC correlations of H-12/C-17 and H_3_-22/C-15. Therefore, the planar stucture of **2** was assigned. The relative stereochemistry of **2** was identical to that of **1** by inspection of their ROESY data (Figure 3). For the relative configuration at C-3, ROESY correlations of H-3/H_3_-20, H_3_-21 are observed (Figure 2), which indicated that H-3 is an equatorial position. Thus, concluding from the molecular model study, there are two possibilities: H-3*β* orientation/chair conformation of ring A and H-3*α* orientation/boat conformation of ring A, while the former is a more stable configuration, thus we deduced that the relative configuration of H-3 is *β*-orientation. In addition, the chemical shift of C-3 in **2** is in accordance with that of such types of structures wherein 3*α*-OH is around 3 ppm upshifted relative to 3*β*-OH, further securing the configuration at C-3 [11,12,13]. To clarify the absolute configuration of **2**, ECD calculations were performed. It was found that the calculated (3*R*,5*R*,8*R*,9*R*,10*R*,14*S*)-**2** matches well with that experimental curve, demonstrating the absolute configuration of **2** is 3*R*,5*R*,8*R*,9*R*,10*R*,14*S* (Figure 4), which confirmed the deduction of the molecular model study. Collectively, the structure of **2** was finally deduced and named as euphraticanoid B.

Compound **3** was obtained as yellow oils with negative optical rotation ([*α*]_D_^25^ −10.94 in MeOH). The molecular formula of **3** was deduced as C_12_H_18_O_2_ derived from its HRESIMS, ^13^C NMR, and DEPT spectra. The ^1^H NMR spectrum of **3** (Table 2) gives signals for two methyl singlets. The ^13^C NMR and DEPT spectra (Table 2) contain 12 carbon signals classified into two methyl, five methylene, one methine, one oxygenated tertiary carbon, two sp^2^ quaternary carbons, and one keto-carbonyl carbon. The structure architecture of **3** was mainly assembled with the aid of 2D NMR experiments. The ^1^H-^1^H COSY spectrum shows spin systems consisting of H-13/H-2/H-3 and H-7/H-8/H-9. The HMBC correlations of H_3_-12/C-3, C-4, C-5 and H-1, H-2/C-5 led to conclude the presence of five-membered ring. Additional HMBC correlations of H-7, H-8/C-6 (*δ*_C_ 202.1), H-8, H-9/C-10 (*δ*_C_ 75.1), H-7/C-5, and H-9/C-1, in combination with the chemical shifts of C-5, C-6, and C-10 allowed to deduce a seven-membered ring as shown (Figure 2). Finally, the HMBC correlations of H_3_-11/C-1, C-9, C-10 and the above mentioned H_3_-12/C-3, C-4, C-5 clarify the positions of two methyl groups. In this way, the planar structure of **3** was identified. The relative configuration of **3** was assigned by ROESY data (Figure 3), which gives a correlation of H-1/H_3_-11, indicating that these protons are adjacent to each other. Finally, the absolute configuration of **3** was established by ECD calculations. It was found that the calculated ECD of (1*S*,10*R*)-**3** (Figure 4) is in accordance with that of experimental one, eventually clarifying the absolute configuration of **3** to be 1*S*,10*R*, with a trivial name euphraticanoid C.

Compound **4**, obtained as a yellow oil with positive optical rotation ([*α*]_D_^25^ +83.91 in MeOH), was found to have a molecular formula of C_12_H_18_O_2_ by analysis of its HRESIMS, ^13^C NMR, and DEPT spectra. The ^1^H NMR spectrum of **4** (Table 2) displays two methyl singlets. The ^13^C NMR and DEPT spectra (Table 2) contain 12 carbon signals ascribed to two methyl, five methylene, one methine, two sp^2^ and one sp^3^ quaternary carbons, and one keto-carbonyl carbon. In the manner of **3**, the structure of **4** was mainly constructed with the assistance of 2D NMR data. The ^1^H-^1^H COSY spectrum of **4** shows correlations of H-1/H-2/H-3 and H-7/H-8/H-9. Starting from two spin systems, the HMBC correlations of H-3/C-4, C-5, H_3_-11/C-3, C-4, C-5, H-1, H-2/C-10, and H-1/C-5, in consideration of the chemical shift of C-1 suggest the presence of a six-membered ring (A) and the substituted groups thereof. In addition, HMBC correlations of H-8, H-9/C-10, H-9/C-5, H-7, H-8/C-6, H-7/C-5, and H_3_-12/C-1, C-9, C-10, in combination with the chemical shift of C-6 (*δ*_C_ 205.8), imply the presence of another six-membered ring (B) which fuses with ring A via the formation of C-5–C-10 (Figure 1). Collectively, the planar structure of **4** was assigned. There are two chiral centers in the molecule. ROESY correlations of H_3_-12, 1-OH (in DMSO-*d*_6_)/H-9a, H-1/H-9b indicate the spacial relationship of one OH group and 10-CH_3_ (Figure 3). To clarify the absolute configuration of **4**, ECD calculations were utilized, which shows that the calculated ECD curve of (1*R*,10*R*)*-***4** matches well with the calculated one, evidently indicating the absolute configuration are 1*R*,10*R*. Meanwhile, the absolute configuration of **4** was confirmed by Mosher’s method [14]. In brief, treatment of **4** with (*R*)- or (*S*)-a-methoxy-atrifluoromethyl phenylacetic acyl chloride (MTPA-Cl) in deuterated pyridine was carried out to acquire the (*S*)-MTPA ester (**4a**) and (*R*)-MTPA ester (**4b**) (Figure 5), respectively. Further analysis of the ^1^H NMR signals of **4a** and **4b** indicates a 10*R* configuration judged from the *Δδ*_H_ values of **4a** and **4b**. As a result, the absolute configuration of **4** was finally confirmed with a trivial name of euphraticanoid D.

The nine known compounds were identified as populeuphroid L (**5**) [7], epimansumbinol (**6**) [12], ergosta-4,6,8(15),22-tetraen-3-one (**7**) [15], mansumbin-13(17)-en-3,16-dione (**8**) [16], 3*α*-acetoxy-mansumbin-13(17)-en-16-one (**9**) [17], 3-epi-δ-amyrin (**10**) [18], α-boswellic acid (**11**) [19], 11α-ethoxy-β-boswellic acid (**12**) [20], acetyl-11-keto-β-boswellic acid (**13**) [21] by a comparison of their spectroscopic data with those reported in the literature. Although compound **12** is a known one, it might be an artefact produced during extraction procedure.

### 2.2. Biological Evaluation

Neuroexcitotoxicity and oxidative stress have been implicated as playing a dominant role in neurodegenerative disorders such as Alzheimer’s disease (AD), ischemic stroke, as well as Parkinson’s disease (PD) [22,23,24,25,26]. In this study, all the compounds isolated from *P. euphratica* resin were applied to detect neuroprotective bioactivities against glutamate-induced excitotoxicity in SH-SY5Y cells and to examine their antioxidative effects against H_2_O_2_ in HT-22 cells. In primary screening, HT-22 cells were pretreated with 20 μM of different compounds, following by 600 μM H_2_O_2_ stimulation for 24 h. Our results show that eight out of thirteen compounds could significantly prevent H_2_O_2_-induced oxidative stress with compounds **4**, **7**–**9**, **12**, and **13** are more potent (Figure 6A). Therefore, compounds **4**, **7**–**9**, **12**, and **13** were submitted to a dose-dependent response experiment. The results show that **4**, **8**, and **9** could dose-dependently protect neural cells from H_2_O_2_-induced oxidative stress injury (Figure 6B,D,E), and **7**, **12**, and **13** possess neuroprotective property against oxidative stress at lower concentrations (10 μM and 20 μM) (Figure 6C,F,G). Of note, neuroprotective effects of **7**, **12**, and **13** reach the maximum at 20 μM and decline at 40 μM. Last but not the least, all the compounds except **8** show no cytotoxicity toward HT-22 cells even at 40 μM (Figure 6D). Although **8** is cytotoxicity against HT-22 cells at 40 μM, it appears that the cytotoxic effect might be negligible. Interestingly, compounds **1**, **2**, **5**, **6**, **8**, and **9** are all octanortriterpenoids. However, the fact that **8** and **9** are active and the other analogs are inactive indicate the importance of α, β-unsaturated ketone to keep the activity. In addition, it was observed that compounds **12** and **13** are active, while in contrast **10** and **11** are inactive, implying that ursane-type rather than oleanane-type triterpenoids might contribute to neuroprotection. Boldly, it was found that compounds **4**, **7**–**9**, and **13** all bear a common α, β-unsaturated ketone, whether such a functional group is essential for keeping the activity needs further exploration.

The neuroprotective properties of all the compounds were also observed in glutamate-induced SH-SY5Y cells. Cells were pretreated with 20 μM of different compounds followed by 10 mM glutamate stimulation for 24 h. The results show that **3**, **4**, **8**–**10** could significantly prevent glutamate-induced excitotoxicity (Figure 7A). Thereafter, **3**, **4**, **8**, and **9** were assessed for a dose-dependent response and revealed that the neuro protection of **4**, **8**, and **9** against glutamate-induced excitotoxicity is dose-dependent (Figure 7B−E). Despite that octanortriterpenoids **8** and **9** are different types of compounds from **3** and **4**, their neuroprotective activities against glutamate-induced excitotoxicity might imply that the presence of an α,β-unsaturated ketone is pivotal for keeping neuroprotection.

## 3. Experimental Section

### 3.1. General Procedures

UV spectra were obtained on a Shimadzu UV-2600 spectrometer (Shimadzu Corporation, Tokyo, Japan). CD spectra were measured on a Chirascan instrument (Agilent Technologies, Santa Clara, CA, USA). NMR spectra were recorded on a Bruker AV-600 spectrometer (Bruker, Karlsruhe, Germany) with TMS as an internal standard. HRESIMS of **1**–**4** was collected by a Shimazu LC-20AD AB SCIEX triple TOF 5600+ MS spectrometer (Shimadzu Corporation, Tokyo, Japan). Column chromatography was undertaken on silica gel (200–300 mesh, Qingdao Marine Chemical Inc., Qingdao, China), MCI gel CHP 20P (75–150 μm, Mitsubishi Chemical Industries, Tokyo, Japan), RP-18 (40–60 µm; Daiso Co., Tokyo, Japan), and Sephadex LH-20 (Amersham Pharmacia, Uppsala, Sweden). Optical rotations were measured on a Bellingham + Stanley ADP 440 + digital polarimeter (Bellingham & Stanley, Kent, UK). Semi-preparative or analytic HPLC was carried out using an Agilent 1200 liquid chromatograph (Agilent Technologies, Santa Clara, CA, USA). The column used was a YMC-Pack ODS-A 250 × 9.4 mm, i.d., 5 µm, or a Phenomenex Kinetex (250 × 10 mm, i.d., 5 μm).

### 3.2. Plant Material

The resins of *P. euphratica* were collected by Ming-Yang Zong from Bayin, Xinjiang Autonomous Region, in November 2011. A voucher specimen (CHYX0573) identified by Prof. Bin Qiu at Yunnan University of Traditional Chinese Medicine is deposited at School of Pharmaceutical Sciences, Shenzhen University, China.

### 3.3. Extraction and Isolation

The dried resins of the title plant (50 kg) was soaked with 95% EtOH (300 L × 3 × 24 h) to afford a crude extract, which was suspended in water and partitioned with EtOAc to afford an EtOAc soluble extract (12 kg). This extract was cut into eight fractions (Fr.1–Fr.8) by using a silica gel column with petroleum ether–acetone (50:1, 35:1, 20:1, 15:1, 10:1, 7:1, 3:1,1:1) as solvents. Fr.1 (640 g) was separated via MCI gel CHP 20P eluted with aqueous MeOH (50%–100%) to provide nine portions (Fr.1.1–Fr.1.9). Among them, Fr.1.5 (136 g) was subjected to a RP-18 column eluted with aqueous MeOH (70%–100%) to provide nine portions (Fr.1.5.1–Fr.1.5.9). Fr.1.5.5 (38.2 g) was further separated via a silica gel column washed with petroleum ether–CH_2_Cl_2_ (30:1, 20:1, 15:1, 13:1, 10:1, 8:1, 4:1, 2:1, 1:1, 0:100) to provide six portions (Fr.1.5.5.1–Fr.1.5.5.6). Fr.1.5.5.4 (5.731 g) was further separated via vacuum liquid chromatography and washed with petroleum ether–EtOAc (30:1) to provide six portions (Fr.1.5.5.4.1–Fr.1.5.5.4.6). Of them, Fr.1.5.5.4.4 (373 mg) were subjected to preparative TLC (petroleum ether-CH_2_Cl_2_ (2:1) to give Fr.1.5.5.4.4.1–Fr.1.5.5.4.4.6. Fr.A.5.5.4.4.3 (9 mg) was submitted to Sephadex LH-20 (MeOH) to afford a portion (8 mg), which was further purified by semi-preparative HPLC with aqueous MeOH (94%) to afford compound **6** (2.64 mg, *t*_R_ = 29.605 min; flow rate: 3 mL/min). Fr.A.5.5.4.4.4 (13 mg) was submitted to Sephadex LH-20 (MeOH) to give a portion (10 mg), which was purified by semi-preparative HPLC (aqueous MeCN, 40%) to give two portions. **1** (1.58 mg, *t*_R_ = 13.575 min; flow rate: 3 mL/min) and **2** (0.88 mg mg, *t*_R_ = 14.728 min; flow rate: 3 mL/min) was obtained from Fr.A.5.5.4.4.4.1 (2.50 mg, *t*_R_ = 28.328 min, flow rate: 3 mL/min) by HPLC separation (aqueous MeOH, 94%). Fr.1.5.5.5 (1.609 g) was further separated via a silica gel column washed with petroleum ether–EtOAc (50:1,25:1,15:1,10:1) to provide seven portions (Fr.1. 5.5.5.1–Fr.1. 5.5.5.7). Fr.1.5.5.5.6 (128 mg) were subjected to preparative TLC (petroleum ether -CH_2_Cl_2_, 2:1) to give Fr.1. 5.5.5.6.1–Fr.1. 5.5.5.6.4. Fr.1.5.8 (721.0 mg) was purified by Sephadex LH-20(MeOH) to afford two parts (Fr.1.5.8.1 and Fr.1.5.8.2). Of which, Fr.1.5.8.2 (240.0 mg) was further divided into six parts (Fr.1.5.8.2.1–Fr.1.5.8.2.6) by a vacuum liquid chromatography eluted with petroleum ether–EtOAC (100:1–1:1). Fr.1.5.8.2.4 (56.9 mg) was separated by semi-preparative HPLC with aqueous MeOH (97%) to afford **7** (8.9 mg, *t*_R_ = 35.6 min, flow rate: 3 mL/min). Fr.1.6 (36 g) was divided into eight parts (Fr.1.6.1–Fr.1.6.8) by using a silica gel column eluted with petroleum ether–acetone (250:1–1:1). Fr.1.6.4 (4.6 g) was submitted to Sephadex LH-20 (MeOH) to yield four fractions (Fr.A.6.4.1–Fr.A.6.4.4). Fr.1.6.4.3 (2.2 g) was subjected to a RP-18 column eluted with aqueous MeOH (65%–100%) to yield five fractions (Fr.1.6.4.3.1–Fr.1.6.4.3.5). **5** (29.4 mg, *t*_R_ = 18.9 min, flow rate: 3 mL/min) was obtained from Fr.1.6.4.3.4 (500.0 mg) by a silica gel column eluted with petroleum ether–acetone (300:1–1:1) and HPLC separation (aqueous MeCN, 82%). Fr.1.7 (60 g) was subjected to a RP-18 column eluted with aqueous MeOH (60%–100%) to yield thirteen fractions (Fr.1.7.1–Fr.1.7.13). Fr.1.7.3 (332 mg) was further separated via Sephadex LH-20 (MeOH) and semi-preparative HPLC with aqueous MeCN (87%) to afford **8** (26.4 mg, *t*_R_ = 12.3 min, flow rate: 3 mL/min). Fr.1.7.4 (193.0 mg) was passed through Sephadex LH-20 (MeOH) to yield two fractions (Fr.1.7.4.1 and Fr.1.7.4.2). Fr.1.7.4.1 (103 mg) was further purified by semi-preparative HPLC with aqueous MeCN (75%) to afford **9** (6.4 mg, *t*_R_ = 18.8 min; flow rate: 3 mL/min). Fr.1.7.13 (33.2 g) was divided into six fractions (Fr.1.7.13.1–Fr.1.7.13.6) by using silica gel chromatography (petroleum ether–acetone, 40:1–1:1). Compound **10** (3.0 g) was purified from Fr.1.7.13.1 (14.0 g) by using gradient silica gel chromatography (petroleum ether–CHCl_3_, 10:1–1:1). Fr. 5 (370 g) was separated via MCI gel CHP 20P eluted with aqueous MeOH (40%–100%) to provide seven portions (Fr.5.1–Fr.5.7). Fr.5.1 (2.354 g) was passed through Sephadex LH-20 (MeOH) to yield two fractions (Fr.5.1.1 and Fr.5.1.2). Fr.5.1.1 (1.036 g) was separated via RP-18 eluted with aqueous MeOH (45%–100%) to provide six portions (Fr.5.1.1.1–Fr.5.1.1.6). Fr.5.1.1.2 (158 mg) was purified by semi-preparative HPLC (aqueous MeCN, 24%) to yield **3** (61.08 mg, *t*_R_ = 21 min; flow rate: 3 mL/min) and three fractions (Fr.5.1.1.2.1–Fr.5.1.1.2.3). Fr.5.1.1.2.3 (21.0 mg) was further purified by semi-preparative HPLC to afford **4** (9.05 mg, *t*_R_ = 24.047 min, flow rate: 3 mL/min) by HPLC separation (aqueous MeOH, 60%). Compounds **11** (9.95 mg, *t*_R_ = 40.028 min; flow rate: 3 mL/min), **12** (3.16 mg, *t*_R_ = 36.118 min; flow rate: 3 mL/min) and **13** (6.65mg, *t*_R_ = 19.938 min; flow rate: 3 mL/min) were afforded from the portion of Fr.5.6 (50.3 mg) by semi-preparative HPLC (aqueous MeOH, 90%).

### 3.4. Compound Characterization Data

Compound **1**: White powder. [α]^20^_D_ +2.0 (*c* 0.10, MeOH); CD (MeOH), *∆ε*_2__02_ +13.70, *∆ε*_226_ −2.99, *∆ε*_280_ +0.30; UV (MeOH) *λ*_max_ (log *ε*) 256 (0.18) nm; HRESIMS: *m*/*z* 315.2676 [M + H]^+^ (calcd. for C_22_H_3__5_O, 315.2682); ^1^H and ^13^C NMR data, see Table 1.

Compound **2**: White powder. [α]^20^_D_ +7.3 (*c* 0.06, MeOH); CD (MeOH), *∆ε*_2__02_ +9.59, *∆ε*_226_ −1.40; UV (MeOH) *λ*_max_ (log *ε*) 238 (0.28) nm; HRESIMS: *m*/*z* 315.2672 [M + H]^+^ (calcd. for C_22_H_3__5_O, 315.2682); ^1^H and ^13^C NMR data, see Table 1

Compound **3**: Yellow syrup. [α]^20^_D_ −10.9 (*c* 0.06, MeOH); CD (MeOH), *∆ε*_201_ −8.59, *∆ε*_211_ −13.47, *∆ε*_2__56_ +10.56, *∆ε*_32__8_ −2.31; UV (MeOH) *λ*_max_ (log *ε*) 257 (1.08) nm; HRESIMS: *m*/*z* 195.1371 [M + H]^+^ (calcd. for C_12_H_1__9_O_2_, 195.1380); ^1^H and ^13^C NMR data, see Table 2.

Compound **4**: Yellow syrup. [α]^20^_D_ +83.9 (*c* 0.09, MeOH); CD (MeOH), *∆ε*_204_ −20.38, *∆ε*_248_ +60.83, *∆ε*_326_ −2.87; UV (MeOH) *λ*_max_ (log *ε*) 248 (1.14) nm; HRESIMS: *m*/*z* 195.1373 [M + H]^+^ (calcd. for C_12_H_1__9_O_2_, 195.1380); ^1^H and ^13^C NMR data, see Table 2.

### 3.5. ECD Calculation for Compounds ***1**–**4***

Conformation search using molecular mechanics calculations was performed in CONFLEX version 7.0 with MMFF force field with an energy window for acceptable conformers (ewindow) of 5 kcal/mol above the ground state, a maximum number of conformations per molecule (maxconfs) of 100, and an RMSD cutoff (rmsd) of 0.5Å. Then the predominant conformers were optimized at B3LYP/6-311+G(d) level in Gaussian 09 [27]. The optimized conformation geometries and thermodynamic parameters of all selected conformations were provided. The optimized conformers of **1**–**4** were used for the ECD calculation, which were performed with Gaussian 09 (B3LYP/6-311+G(d)). The solvent effects were taken into account by the polarizable-conductor calculation model (PCM, methanol as the solvent). Percentages for each conformation are shown in Appendix A.

### 3.6. MTPA Esterification of ***4***

Compound **4** (1 mg) was dissolved in 600 μL of anhydrous deuteration pyridine, which was divided into two equal portions in NMR sample tube. To each portion was added 1.5 μL of either *R*-MTPA-Cl or *S*-MTPA-Cl, and then the mixtures were kept at room temperature for 2 h. Finally, the ^1^H NMR data were collected using the mixtures without purification.

### 3.7. Bioactivity Assay

Mouse hippocampus cell line (HT-22 cells) was purchased from iCell company in Shanghai, China, and human neuroblastoma cell line (SH-SY5Y) was purchased from Cellcook Company in Guangzhou, China. The experimental procedures of cell culture and treatments were performed as described previously [28]. Briefly, cells were cultured in Dulbecco’s modified Eagle’s medium (DMEM, Gibco, Grand Island, NY, USA) with the addition of 10% fetal bovine serum (FBS, Gibco, Australia), 100 U/mL penicillin and 100 U/mL streptomycin (Gibco, Thornton, Australia) at 37 °C in a humidified atmosphere of 95% air and 5% CO_2_ incubator. SH-SY5Y cells were plated at a density of 2 × 10^4/^well of a 96-well plate for 24 h. Thereafter, SH-SY5Y cells were pretreated with compounds (10, 20 and 40 μM) or PBS, and followed by a 24 h stimulation of either 10 mM glutamate (Sigma) or PBS (HyClone). HT-22 cells were pretreated with compounds (10, 20 and 40 μM) or PBS and followed by a 24 h stimulation of either 600 μM H_2_O_2_ (Sigma) or PBS. After all treatments, 10 μL/well of CCK-8 solution was added into each well and incubated for 1 h. The absorbance was determined at 450 nm by using a microplate reader (BioTek, Winooski, VT, USA).

All quantified biological data are expressed as means ± standard error of the mean (SEM) of n independent experiments. Statistical analyses were performed by one-way analysis of variance (ANOVA) following by a post-hoc multiple-comparison Tukey test whereby *p* < 0.05 (** *p* < 0.01, *** *p* < 0.001) was considered significant.

## 4. Conclusions

To conclude, the current study led to the characterization of two new nortriterpenoids (**1** and **2**), two new norsesquiterpenoids (**3** and **4**), and nine known compounds (**5**–**13**) from *P. euphratica* resins. Biological evaluation revealed that **4**, **7**–**9**, **12**, and **13** are neuroprotective agents and the presence of α,β-unsaturated ketone in the structure might be crucial for keeping the activity. This study might shed light on further structure modification for developing new generation of neuroprotective drugs.

## Figures and Tables

**Figure 1 molecules-24-04379-f001:**
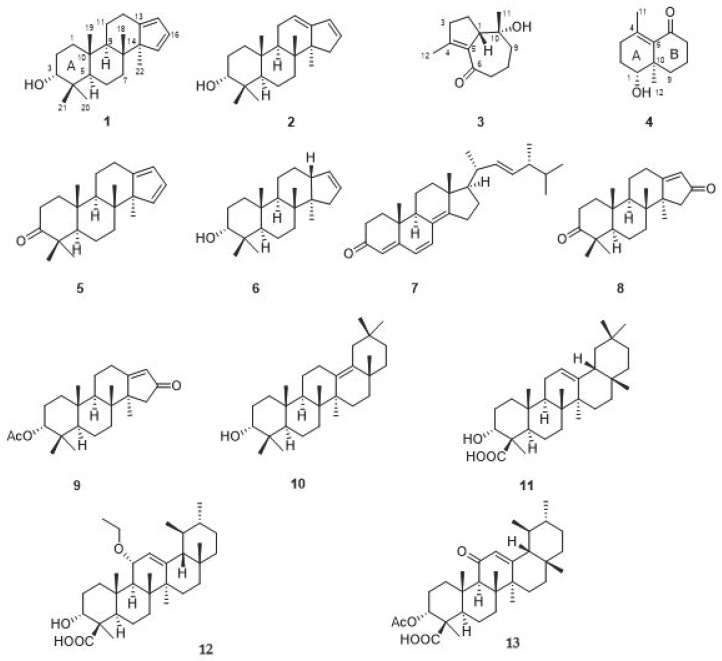
The structures of compounds **1**–**13**.

**Figure 2 molecules-24-04379-f002:**
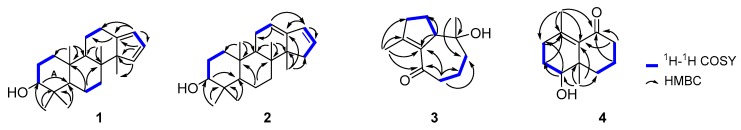
Key ^1^H−^1^H COSY and HMBC correlations for **1**–**4**.

**Figure 3 molecules-24-04379-f003:**
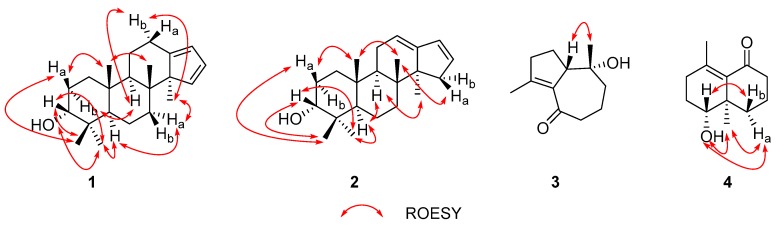
Key ROESY correlations for **1**–**4**.

**Figure 4 molecules-24-04379-f004:**
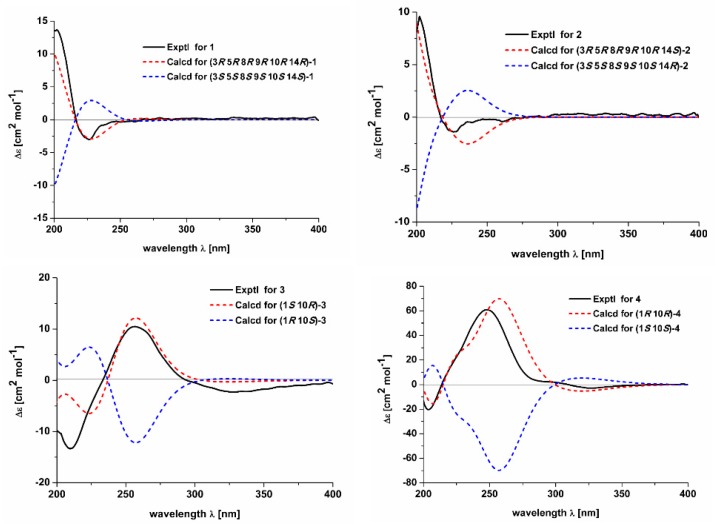
The calculated and experimental ECD spectra of **1**–**4**.

**Figure 5 molecules-24-04379-f005:**
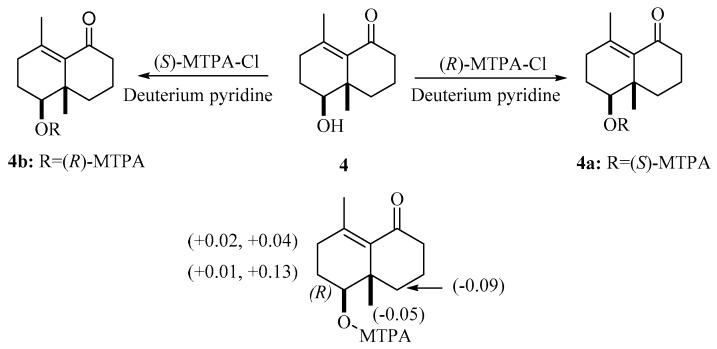
*Δδ*_H_ values of the Mosher esters **4a** and **4b**.

**Figure 6 molecules-24-04379-f006:**
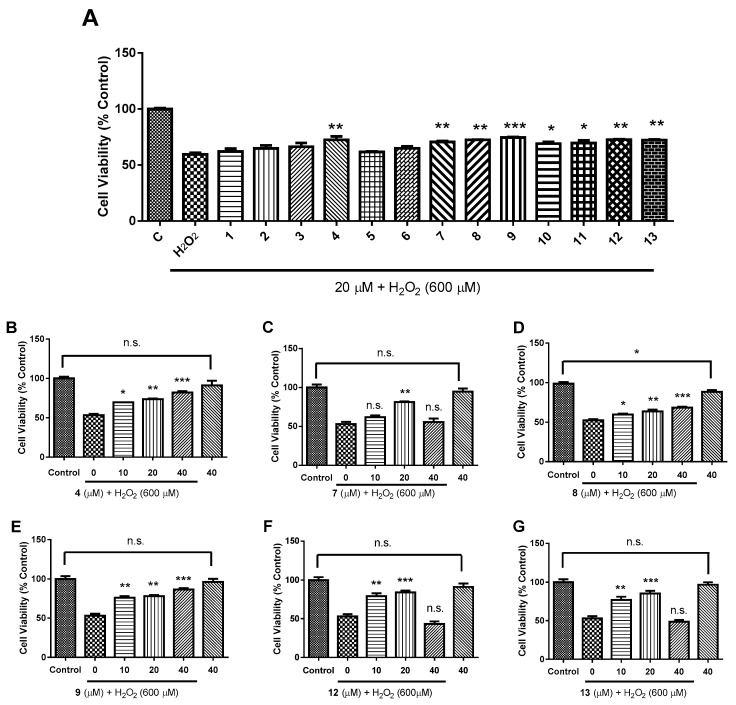
Neuroprotective activities of compounds **1**−**13** against H_2_O_2_-mediated oxidative stress in HT-22 cells. Cell viability was determined by CCK-8 assay. (**A**): Primary screening at 20 μM; (**B**–**G**): Dose-dependent curve of **4**, **7**–**9**, **12**, and **13**. Control was PBS-treated cells. “40” in bar charts means that cells were only treated by 40 μM compound. *n* = 3, all data in bar charts represent means ± SEM. The symbol n.s. means no significance, * *p* < 0.05, ** *p* < 0.01, *** *p* < 0.001, one-way ANOVA with post-hoc comparison Turkey.

**Figure 7 molecules-24-04379-f007:**
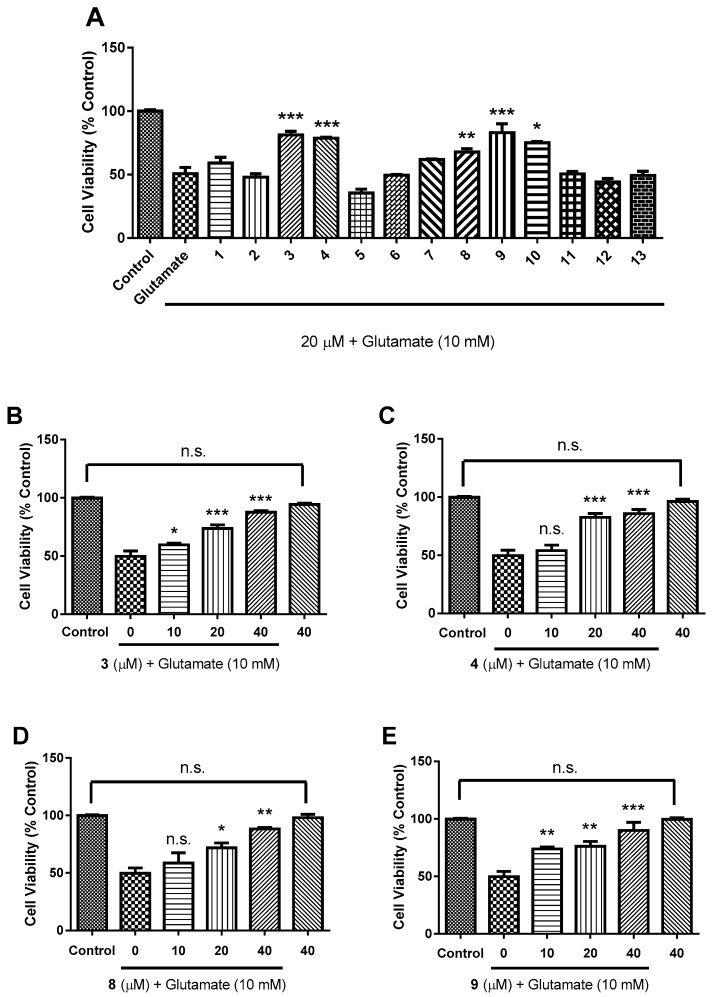
Neuroprotective activities of compounds **1**–**13** against glutamate-induced excitotoxicity in SH-SY5Y cells. Cell viability was determined by CCK-8 assay. (**A**): Primary screening at 20 μM; (**B**–**E**): the neuroprotection of **3**, **4**, **8** and **9** was dose-dependent. Control was PBS-treated cells. “40” in bar charts means that cells were only treated by 40 μM compound. *n* = 3, all data in bar charts represent means ± SEM. The symbol n.s. means no significance, * *p* < 0.05, ** *p* < 0.01, *** *p* < 0.001, one-way ANOVA with post-hoc comparison Turkey.

**Table 1 molecules-24-04379-t001:** ^1^H (600 MHz) and ^13^C (150 MHz) NMR data of **1** and **2** in CDCl_3._

	1			2	
no.	*δ* _H_	*δ* _C_	no.	*δ* _H_	*δ* _C_
1	Ha: 1.50 (m)	34.2, CH_2_	1	Ha: 1.38 (m)	33.2, CH_2_
	Hb: 1.40 (m)			Hb: 1.33 (overlap)	
2	Ha: 1.97 (m)	25.6, CH_2_	2	Ha: 1.96 (m)	25.3, CH_2_
	Hb: 1.57 (m)			Hb: 1.56 (m)	
3	3.42 (t-like, 2.63)	76.3, CH	3	3.42 (t-like, 2.8)	76.3, CH
4		37.8, C	4		37.6, C
5	1.36 (m)	49.7, CH	5	1.33 (overlap)	49.7, CH
6	Ha: 1.48 (m)	18.4, CH_2_	6	Ha: 1.48 (m)	18.3, CH_2_
	Hb: 1.38 (m)			Hb: 1.44 (m)	
7	Ha: 1.84 (m)	37.3, CH_2_	7	Ha: 1.62 (m)	34.5, CH_2_
	Hb: 1.51 (overlap)			Hb: 1.31 (m)	
8		41.4, C	8		37.6, C
9	1.51 (overlap)	51.2, CH	9	1.83 (dd, 10.6, 7.1)	48.2, CH
10		37.9, C	10		37.5, C
11	Ha: 1.70 (m)	23.1, CH_2_	11	Ha: 2.13 (m)	23.6, CH_2_
	Hb: 1.34 (m)			Hb: 1.93 (m)	
12	Ha: 2.61 (ddd, 13.6, 4.7, 1.7)	26.4, CH_2_	12	5.40 (t, 3.7)	114.7, CH
	Hb: 2.16 (tdd, 13.6, 5.2, 1.7)				
13		156.7, C	13		151.9, C
14		61.4, C	14		49.4, C
15	6.16 (d, 5.4)	142.7, CH	15	Ha: 2.59 (brd,17.4)	41.4, CH_2_
				Hb: 1.85 (brdt,17.4, 1.8)	
16	6.21 (dd, 5.4, 1.7)	129.6, CH	16	5.83 (dt, 5.7, 2.6)	133.5, CH
17	5.79 (q-like, 1.7)	120.2, CH	17	6.04 (dt, 5.7, 1.8)	131.3, CH
18	0.64 (s)	15.5, CH_3_	18	0.71 (s)	17.6, CH_3_
19	0.84 (s)	16.3, CH_3_	19	0.93 (s)	15.2, CH_3_
20	0.98 (s)	28.5, CH_3_	20	0.97 (s)	28.5, CH_3_
21	0.84 (s)	22.4, CH_3_	21	0.86 (s)	22.3, CH_3_
22	1.05 (s)	17.3, CH_3_	22	1.08 (s)	24.9, CH_3_

**Table 2 molecules-24-04379-t002:** ^1^H (600 MHz) and ^13^C (150 MHz) NMR data of **3** and **4** in CDCl_3_.

	3			4	
no.	*δ* _H_	*δ* _C_	no.	*δ* _H_	*δ* _C_
1	3.18 (m)	56.7, CH	1	3.61 (dd, 11.4, 4.6)	76.6, CH
2	Ha: 1.99 (m)	23.7, CH_2_	2	1.72 (m)	26.7, CH_2_
	Hb: 1.93 (m)				
3	Ha: 2.45 (m)	39.0, CH_2_	3	2.19 (m)	32.4, CH_2_
	Hb: 2.36 (m)				
4		158.3, C	4		139.0, C
5		135.3, C	5		138.2, C
6		202.1, C	6		205.8, C
7	2.46 (m)	45.2, CH_2_	7	Ha: 2.49 (m)	43.2, CH_2_
				Hb: 2.29 (m)	
8	Ha: 1.86 (m)	21.4, CH_2_	8	Ha: 1.94 (m)	20.8, CH_2_
	Hb:1.49 (m)			Hb: 1.90 (m)	
9	Ha: 1.95 (m)	46.8, CH_2_	9	Ha: 2.09 (ddd, 13.2, 5.2, 3.3)	36.7, CH_2_
	Hb: 1.72(m)			Hb: 1.53 (ddd, 13.2, 11.1, 6.4)	
10		75.1, C	10		42.7, C
11	0.99 (s)	20.8, CH_3_	11	1.74 (s)	21.2, CH_3_
12	2.07 (brs)	17.2, CH_3_	12	0.81 (s)	18.8, CH_3_
10 -OH	4.83 (s) ^a^		1 -OH	4.67 (d, 4.8) ^a^	

^a^ In DMSO−*d*_6_.

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
