# Peer review of "Neuroprotective Norsesquiterpenoids and Triterpenoids from Populus euphratica Resins"

_molecules, 2019, doi:10.3390/molecules24234379_

Round 1

Reviewer 1 Report

The present manuscript was aimed to investigate the chemical constituents from the ethanol extracts of P. euphratica resins. Totally twelve compounds including four new ones were characterized and their structures were determined by spectrocsopic, chemical, and computational methods. In addition, the neuroprotective bioactivities of these compounds were examined and the SAR was also discussed. The English language and style of this manuscript were edited well, and the structural elucidations were established in a systematic manner. Therefore, this manuscript is recommended to accept for publication in Molecules after some revisions addressed as following.

There were still some minor typographic, grammar, and format errors to be observed in the manuscript. Authors have to check and revise these errors. For examples, lines 61-62, the sentence was repeated; Table 2, some minor mistakes; Section 3.1, some more digits should be removed, etc. The reported significant figures should be uniform or follow the guidelines of this journal. At least the optical rotation should be presented in one decimal digit only. Regarding the optical rotation data, compounds 1-3 showed very low values. It indicated the presence of stereomeric mixture. Line 78, “five non-protonated carbons (four quaternary)” is not so meaningful. All five carbons should be quaternary. The physical states of compounds 3 and 4 should be revised as “syrup” rather than “oil”. The HRMS results of compounds 1-4 were deviated from the theoretical values too much. These experiments should be performed again to get better data. In the References section, the writing manner of references did not follow the style of this journal. Authors have to check and revise these errors carefully, ex. ref 5, 13, and 20.

Reviewer 2 Report

This is an Interesting paper where the authors find new Norsesquiterpenoids and Triterpenoids some of them show neuroprotective activities.

From my point of view, only minor changes should be made.

In line 31 a space is missing between tuberculos and edadenitis In line 84 NOESY has to be referenced to the figure that represent it (should be figure 3) In line 109 they also have to ad figure 3 as a reference to ROESY. In line 124 after C5, C10 a reference to figure 1 is missing In line 125, after ROESY assay a reference to figure 3 is missing, In line 155 a space is missing between neuro and excitotoxicity. In line 154, in point 2.2 authors state that compounds 4,7-9 and 13 are the most powerful and these compounds are used latter to carry out the dose-dependent response experiments. But compound 10 and 11 also show a significant effect as neurotoxicity protectors (figure 6A) why these compounds were not taken into account in dose-dependent assays? In line 166 a reference to fig 6C and G is missing after “20mM” In line 169 a reference to figure 6D is missing after “40mM” In line 185 an “in” is missing between observed and glutamate. In line 188and 189, authors state that compounds 3,4, 8 and 9 are tested to check that they have a dose-dependent response against glutamate. As can be seen in figure 7A the compound 10 also shows good properties. How come the authors do not test this compound a long with 3,4,8 and 9 in dose-dependent response experiments? What is the meaning of “control” in all the figures? This should be explained somewhere in the paper. A similar situation is found with the “40” that appears in all figures. And the most important question is what is the reason for the authors to carry out only one type of test the neuroprotective effect? Some different testing should be carry out such as FACs, MTT, Apoptosis test…etc…In order to check cellular viability. Supplementary figures references are missing, these references should be added clearly in the text.

Round 2

Reviewer 1 Report

I had gone through the revised manuscript carefully and this manuscript was recommended for acceptance since authors had provided suitable explanation for most of the previous queries.

This manuscript is a resubmission of an earlier submission. The following is a list of the peer review reports and author responses from that submission.